# Stepwise Glucoheptoamidation of Poly(Amidoamine) Dendrimer G3 to Tune Physicochemical Properties of the Potential Drug Carrier: In Vitro Tests for Cytisine Conjugates

**DOI:** 10.3390/pharmaceutics12050473

**Published:** 2020-05-22

**Authors:** Anna Czerniecka-Kubicka, Piotr Tutka, Marek Pyda, Małgorzata Walczak, Łukasz Uram, Maria Misiorek, Ewelina Chmiel, Stanisław Wołowiec

**Affiliations:** 1Faculty of Medicine, University of Rzeszów, 35-310 Rzeszów, Poland; anna_czerniecka@poczta.fm (A.C.-K.); tutka@umlub.pl (P.T.); 2National Drug and Alcohol Research Centre, University of New South Wales, Sydney, NSW 2031, Australia; 3Faculty of Chemistry, Rzeszów University of Technology, 35-939 Rzeszów, Poland; mpyda@utk.edu (M.P.); mwalczak@prz.edu.pl (M.W.); luram@prz.edu.pl (Ł.U.); mczygier@prz.edu.pl (M.M.); ewelinachmiel@prz.edu.pl (E.C.); 4Department of Biophysics, Poznań University of Medical Sciences, 60-780 Poznań, Poland

**Keywords:** PAMAM G3 dendrimer, glucoheptoamidation, dynamic light scattering, differential scanning calorimetry, cytisine conjugate, cytotoxicity, internalization, BJ normal human fibroblasts

## Abstract

Third-generation poly(amidoamine) dendrimer (PAMAM) was modified by stepwise primary amine group amidation with d-glucoheptono-1,4-lactone. The physicochemical properties of the conjugates—size, ζ potential in lysosomal pH 5 and in neutral aqueous solutions, as well as intramolecular dynamics by differential scanning calorimetry—were determined. Internalization and toxicity of the conjugates against normal human fibroblasts BJ were monitored in vitro in order to select an appropriate carrier for a drug delivery system. It was found that initial glucoheptoamidation (up to 1/3 of amine groups of neat dendrimers available) resulted in increase of conjugate size and ζ potential. Native or low substituted dendrimer conjugates accumulated efficiently in fibroblast cells at nontoxic 1 µM concentration. Further substitution of dendrimer caused consistent decrease of size and ζ potential, cell accumulation, and toxicity. All dendrimers are amorphous at 36.6 °C as determined by differential scanning calorimetry (DSC). The optimized dendrimer, half-filled with glucoheptoamide substituents, was applied as carrier bearing two covalently attached cytisine molecules: a rigid and hydrophobic alkaloid. The conjugate with 2 cytisine and 16 glucoheptoamide substituents showed fast accumulation and no toxicity up to 200 µM concentration. The half-glucoheptoamidated PAMAM dendrimer was selected as a promising anticancer drug carrier for further applications.

## 1. Introduction

Polyamidoamine dendrimers (PAMAM) have been tested as carriers for drug delivery, drug encapsulation, and gene delivery since their discovery by Tomalia et al. in 1985 [1]. Due to their cationic character, they associate with negatively charged membranes in biological systems, permeate mostly in an endocytic way and reveal enhanced permeation and retention effects [2]. However, amine-terminated PAMAM dendrimers are hemotoxic [2,3]. The hemotoxicity of PAMAM generation 3–5 dendrimers (G3–G5) increases with generation and precludes their straightforward application as drug carriers bearing either absorbed (encapsulated) or covalently attached pro-drugs to the PAMAM core. There is a variety of PAMAM terminal amine group modifications reported till now, like PEG-ylation, acylation, and hydroxyalkylation [3,4,5,6,7,8,9,10,11,12], which modify not only the basicity of intrinsic nitrogen atoms but also tune surface hydrophilicity vs. hydrophobicity. On the other hand, amine groups are convenient peripheral sites to attach potential ligands like folate or biotin and are responsible for binding a conjugated carrier selectively to cancer cell membranes [5,8,10,13,14,15,16,17,18]. Conjugates bearing anticancer drugs, targeting molecules, and surface-modifying substituents may serve as selective and systematically nontoxic anticancer drugs. Such conjugates must fulfill the following criteria: (i) high affinity to cell membrane, (ii) good membrane permeation, (iii) water solubility for intravenous administration, (iv) selective binding to cancer cell membrane, (v) low systemic toxicity, (vi) prolonged blood circulation half-time, and (vii) ability to release a drug from a conjugate. The latter criterion requires cleavable linker. PAMAM dendrimer-based conjugates are good candidates as multidrug carriers to challenge against multidrug resistant cancers [19,20].

Here, we have prepared series of PAMAM G3 dendrimers modified by stepwise substitution of primary amine groups by amide bonds in reaction with d-glucoheptono-1,4-lactone, which was previously used to eradicate amine groups [9,10]. Such substitution allows to provide six hydroxyl groups per one glucoheptoamide substituent and to keep low molecular weight dispersity contrary to commonly used PEG-ylation [4] as well as to maintain very good water solubility of modified PAMAM [9,10,11]. We examined partially glucoheptoamidated G3 by monitoring the molecular size, ζ potential, electrophoretic mobility, and elasticity of the conjugates by differential scanning calorimetry in order to tune the conjugate physical properties. We also used a chosen carrier to covalently bind cytisine, the alkaloid used in antinicotinic therapy [21], and included the cytisine conjugates in our physical studies. Finally, the conjugates were tested for permeation and toxicity on human cell lines.

## 2. Materials and Methods

### 2.1. Reagents and Methods

All the chemicals used in synthesis of PAMAM G3 dendrimer and its conjugates were purchased from SigmaAldrich, except cytisine, which was kindly provided by ADAMED PHARMA S.A. (Pieńków, Poland).

Normal human skin fibroblasts (BJ), Eagle’s Minimum Essential Medium (EMEM), fetal bovine serum (FBS), penicillin, and streptomycin solution were obtained from American Type Culture Collection (ATCC, Manassas, VA, USA). Trypsin-EDTA solution, phosphate-buffered saline (PBS) with and without magnesium and calcium ions, 0.4% trypan blue solution, neutral red, and other chemicals and buffers were provided by Sigma-Aldrich (St Louis, MO, USA). DAPI (4′,6-diamidino-2-phenylindole, dihydrochloride) was purchased from Thermo Fischer Scientific (Waltham, Massachusetts, USA). Cell culture plates/flasks were from Corning Incorporated (Corning, NY, USA) or Nunc (Roskilde, Denmark).

#### 2.1.1. Spectroscopy

IR spectra were recorded with ALPHA FT-IR Bruker instrument in ATR mode or in KBr pellets. The NMR spectra were recorded with Bruker 300 MHz instrument (Rheinstetten, Germany). Spectral assignments were performed based on 2-D ^1^H-^1^H COSY and ^1^H-^13^C HSQC and HMBC experiments.

#### 2.1.2. Differential Scanning Calorimetry

Thermal analysis of G3^gh^ and G3^Cgh^ conjugates and derivatives was carried out in the temperature range of −50 °C to 100 °C using the differential scanning calorimetry (DSC) of a TA 2920 from TA Instruments, Inc. (New Castle, DE, USA). The applied DSC was the heat-flux type calorimeter and was connected with a mechanical refrigerator to control heating and cooling of the sample. All experiments were performed in a nitrogen atmosphere with a constant flow rate of around 50 mL/min. The masses of the samples used for measurements were about 10 mg. The measurements of experimental heat-flow rates were obtained at a heating rate of 10 °C/min after previous cooling at 10 °C/min by standard DSC. The heat-flow rate data were collected from the second heating run after controlled cooling. The temperature and heat-flow rate calibration in the DSC apparatus were performed using parameters of melting indium (T_m_(onset) = 156.6 °C, ΔH_f_ = 28.45 J/g (3.281 kJ/mol)) In order to obtain accurate results of the changes of heat capacity at the glass transition temperature, the calibration of heat capacity was done with a sample of sapphire.

#### 2.1.3. Conjugate Size and ζ Potential

Dynamic light scattering and ζ potential for G3^gh^ conjugates were measured at pH 5 (0.04 M acetate buffer) and in water using Zetasizer Nano instrument (Malvern, UK) for ca. 1 mg/mL samples (ca. 0.1 mM solutions).

#### 2.1.4. Electrophoresis

The electrophoretic mobility of conjugates labeled with fluorescein was studied by polyacrylamide gel electrophoresis. Continuous native PAGE under acidic or basic conditions was performed with vertical Mini-PROTEAN Tetra Cell electrophoresis system (Bio-Rad, Hercules, CA, USA) using Minigels (100 mm × 8 mm × 1 mm). Two types of buffers were used: 80 mM β-alanine-acetate, pH 5.0, and 50 mM Tris-acetate, pH 7.2. 2 µµL of a sample containing 1 µL dendrimers (10 µg) and 1 µL charge buffer (30% glycerol, 8 mM β-alanine-acetate, pH 5.0, or 5 mM Tris-acetate, pH 7.2) was applied to 4% gel. The electrophoresis was performed at 200 V at room temperature for about 30 min under acidic conditions and 20 min under alkaline conditions.

### 2.2. Chemical Syntheses

#### 2.2.1. PAMAM G3 Substituted with Glucoheptonamide

PAMAM G3 dendrimer was obtained at 5 milimolar scale according to the procedure published by Tomalia et al. [1] and stored as 20.1 mM solution in methanol for further use. Then, G3 was substituted with d-glucoheptono-1,4-lactone (GHL) by the addition of solid **GHL** (50, 100, 150, and 200 mg) into four test tubes containing 278 mg G3 (41.2 µmol) in 3 mL DMSO each, stirred until **GHL** dissolved, and heated at 50 °C for 12 h. In separate synthesis, the **G3^32gh^** compound was obtained according to the same protocol except using 20% stoichiometric excess of **GHL** as described before [12].

Then, the solutions were transferred into dialytic tube (nitrocellulose, MW_cutoff_—3.5 kD) and dialyzed against water for 2 days (5 times 3 dm^3^). Water was evaporated under reduced pressure, and products were identified by ^1^H-NMR spectroscopy as G3 was substituted with averages of 5, 10, 16, and 18 equiv. of gh per one PAMAM G3 molecule: **G3^5gh^**, **G3^10gh^**, **G3^16gh^**, and **G3^18gh^**, respectively. The isolated yield was above 80% in every case. The ^1^H-NMR spectra of starting **G3** and synthesized **G3^5gh^**, **G3^10gh^**, **G3^16gh^**, **G3^18gh^**, and **G3^32gh^** are presented in Figure 1.

#### 2.2.2. Conjugates of G3^gh^ with Cytisine

In the first step, cytisine (**C**) was derivitized into *N*-(*p*-nitrophenylcarbonate) as follows: 188 mg C (1.72 mmol) was dissolved in 5 mL CHCl_3_ and 0.300 mL of triethylamine was added, followed by addition of 422 mg of *p*-nitrophenylchloroformate (**NPCF**, 20.07 mmol) in portions with vigorous stirring. The mixture was stored at 50 °C for 4 h, and then, chloroform solution of the mixture (20 mL) was triple extracted with water (60 mL). Chloroform phase was separated, the solvent was removed on rotary evaporator, and the resin residue dried overnight under 1 mbar pressure. The mixture was chromatographed on silica gel with chloroform (**C** was selectively eluted) and then with chloroform: ethyl acetate 1:1 *v/v*. From the latter fraction, the solvent was removed on rotavap and a solid residue was dried overnight under reduced pressure. The product was cytisine-*N*-(*p*-nitrophenylcarbonate), (**CNPC**) identified by NMR spectroscopy (Figure 2 and Figure 3). Isolated yield was 193 mg (0.701 mmol), 41%.

**CNPC** was dissolved in 10.0 mL of DMSO (70 mM stock solution) and used to obtain conjugates with G3 of variable amount of C equivalents per G3, containing 2, 3, and 9 equiv. of carbonyl-linked C as follows: (i) 5.0 mL of **CNPC** solution (350 µmol) was added dropwise into 1200 mg G3 (173 µmol) in 5 mL DMSO; (ii) 2.0 mL of **CNPC** solution (140 µmol) was added into 320 mg G3 (46 µmol) in 2 mL DMSO; and (iii) 3.0 mL of **NPCF** solution (210 µmol) was added dropwise into 145 mg G3 (21 µmol) in 3 mL DMSO with vigorous stirring. All the solutions were heated for 12 h at 50 °C. Then, the solution (i) was divided into three equal portions containing 58 µmols of converted G3. To these aliquots, solid **GHL** was added with vigorous stirring: (ia) 193 mg (928 µmol, 16 equiv.); (ib) 277 mg (1334 µmol, 23 equiv.); and (ic) 386 mg (1856 µmol, 32 equiv.). On the other hand, 143 mg (690 µmol, 15 equiv.) and 53 mg (254 µmol, 12 equiv.) of **CNPC** were added to solutions (ii) and (iii), respectively. All five solutions were kept for 12 h at 50 °C. Then, they were transferred into dialytic tube (MW_cutoff_ 3.5 kD) and dialyzed against water for 3 days (6 times 3 dm^3^). Water was evaporated under reduced pressure, and products were identified by ^1^H-NMR spectroscopy as **G3^2C16gh^**, **G3^2C23gh^**, **G3^2C26gh^**, **G3^3C14gh^**, and **G3^9C11gh^** obtained in synthetic routes (ia), (ib), (ic), (ii), and (iii), respectively. Corresponding ^1^H-NMR spectra of synthesized **G3^2C16gh^**, **G3^2C23gh^**, **G3^2C26gh^**, **G3^3C14gh^**, and **G3^9C11gh^** are presented in Figure 4B–F, respectively, in comparison with the spectrum of **C** in CDCl_3_ (trace A).

#### 2.2.3. Conjugates Labeled with FITC

The selected conjugates were fluorescent labeled by stepwise addition of fluorescein isothiocyanate (FITC) dissolved in ethanol into typically 10 µmoles of the following conjugates: **G3**, **G3^5gh^**, **G3^10gh^**, **G3^16gh^**, and **G3^9C11gh^**, and **G3^2CF16gh^** in methanol to obtain **G3^F^**, **G3^5ghF^**, **G3^10ghF^**, **G3^16ghF^**, **G3^9C11ghF^**, and **G3^2C16ghF^** for biological tests and electrophoretic experiments. Each conjugate molecule contained an average one FITC residue.

### 2.3. Biological Tests

#### 2.3.1. Cell Culture

Normal human skin fibroblasts BJ (ATCC CRL-2522) with doubling time ~1.9 day were cultured in EMEM containing 10% heat-inactivated FBS, 100 U/mL penicillin, and 100 µg/mL streptomycin at 37 °C in humidified 95% air with 5% CO_2_. Medium was changed every 2–3 days, and cells were passaged at about 80% confluence with 0.25% trypsin-0.03% EDTA in PBS (calcium and magnesium free). Cell morphology was checked under the Nikon TE2000S Inverted Microscope (Tokyo, Japan) with phase contrast. Viability and cell density were estimated with the Automatic Cell Counter TC20™ (BioRad Laboratories, Hercules, CA, USA), following trypan blue exclusion test.

#### 2.3.2. Toxicity of Conjugates

Influence of **G3**, **G3^5gh^**, **G3^10gh^**, **G3^16gh^**, **G3^2C16gh^**, and **G3^9C11gh^** on normal human skin fibroblasts (BJ cell line) viability was assessed with neutral red assay. BJ cells were seeded into 96-well plate at density of 8 × 10^4^ cells and incubated for 24 h at 37 °C. Next, 2 mM water stock solutions (sterilized by passing through 0.22-µm syringe filters) were diluted in culture medium containing 10% FBS. Working solutions of dendrimers in range of 3.125–200 µM concentrations were added, and following 24 h incubation, neutral red assay was performed as described before [10].

#### 2.3.3. Cellular Internalization

**Time dependent cellular uptake** of studied dendrimers labelled with one molecule of fluorescein isothiocyanate (FITC) was performed with fluorescence microplate reader. The BJ cells were seeded into 96-well black plate at density of 1.2 × 10^4^ cells (full confluence) and incubated 24 h to reach adhesion to the bottom of the well. Next, sterile 5 mM water stock solutions of FITC-labelled conjugates analogs (**G3^F^**, **G3^5ghF^**, **G3^10ghF^**, **G3^16ghF^**, **G3^2C16ghF^**, and **G3^9C11ghF^**) were diluted in culture medium containing 10% FBS. Cells were treated with the nontoxic, 1 µM working solutions of dendrimers for 1, 3, 5, or 24 h. After three times washing with 1× PBS, plates were read with the Infinite M200 PRO Multimode Microplate Reader (TECAN Group Ltd., Switzerland) at 485/535 nm against blank (wells with dendrimers working solutions).

**Confocal microscopy visualization.** Cells were seeded on sterile cover glasses placed in 24-well plates at a density of 6 × 10^4^ cells per well and incubated for 24 h. Cells were treated with dendrimers as described above for 24 h. After 3 times washing with PBS, cells were fixed in 3.7% formalin (for 10 min at room temperature) and then labelled with 600 nM working solution of DAPI. Cover glasses with cells were mounted on glass slide with mounting medium. Specimens were visualized with confocal microscope (Olympus FV10i, Tokyo, Japan) at 488/530 nm for FITC and 405/461 nm for DAPI with 60× water magnification lens. Images were collected in the *Z*-axis position at the largest nuclear cross-sectional area. Pinhole was set for 1.5 airy unit (optical section thickness of approximately 1.5 µm). All images were collected at the same laser power and gain.

#### 2.3.4. Statistical Analysis

For the cell culture assays, to estimate the differences between treated and non-treated control samples, a statistical analysis was performed using the nonparametric Kruskal–Wallis test. *p* < 0.05 was considered as statistically significant. Calculations were performed using Statistica 13.3 software (StatSoft).

## 3. Results and Discussion

PAMAM G3 is a polyelectrolyte of high buffering capacity. The pK_a_ values for amine groups of G0-G5 were determined experimentally by potentiometric titration [22,23]. It was shown that pK_a_ values for primary and tertiary amine groups are ca. 9.0 and 5.8, respectively. Thus, the **G3** dendrimer which provides 62 nitrogen proton acceptors (32 primary and 30 tertiary) is loaded nearly 60^+^ at pH < 5. Generally, at pH 7, tertiary amines are mostly deprotonated, whereas primary amines are protonated, which renders G3 average 32^+^ cation.

It was also demonstrated by potentiometric titration of OH-terminated G4 that primary amine group exclusion results in considerable decrease of pK_a_ of intrinsic tertiary nitrogen centers from 9.2 into 6.3 [22]. Thus, peripheral amine group modification considerably influences the physicochemical properties of entire conjugates. We hypothesized that partial substitution of primary amine groups might modify the properties of entire molecules. Thus, we have used d-glucoheptono-1,4-lactone (**GHL**) to obtain series of partially substituted PAMAM G3 derivatives, as shown in Scheme 1.

### 3.1. Syntheses and Characterization of Conjugates

#### 3.1.1. Glucoheptoamidated PAMAM G3 Dendrimers

PAMAM G3 dendrimer has 32 primary amine groups. They were used for reaction with **GHL** to obtain partially glucoheptoamidated G3 derivatives, which were characterized by ^1^H-NMR spectroscopy. The ^1^H-NMR spectra are presented in Figure 1. The **G3** core methylene proton resonances cover the 2.15–3.40 ppm region with total integration intensity corresponding to [484H]. All shell methylene protons b (next to amide carbonyls) resonate at ca. 2.25 ppm, while all shell methylene protons a (next to branching nitrogen atom) resonate at ca. 2.65 to give two broad signals of [120H] integral intensity. The inner shell resonances of c and d methylene protons are observed at 3.29 and 2.43 ppm with [56H] intensity, while outer shell c_3_ and d_3_ chemical shift depends on outermost nitrogen substituents and are observed at 3.25 and 2.75 ppm in amine-terminated G3 with integral intensity of [64H] both (trace A). In the case of **G3^32gh^** in which all terminal nitrogen atoms are glucoheptoamidated, the c_3_ and d_3_ resonances shift down to ca. 3.3 ppm while d_2_, d_1_, and d_0_ resonances remain within 2.43 ppm region (spectrum not shown here [10]). For partially glucoheptoamidated G3, the integral intensity of d_3′_ resonance at 3.3 ppm grows with the number of gh substituents with concomitant gradual shift of d_3_ resonance of unsubstituted arm down to 2.70 ppm. All the resonances of G3 core as well as gh CHOH proton multiplets are broad due to slow dynamic behavior of partially substituted **G3^gh^** conjugates. Nevertheless, the integration of resonances within 4.20–3.42 ppm and division of the integral by 7 (number of CHOH protons in gh substituent) allowed to estimate the average number of gh substituents in the conjugates, i.e., 5, 10, 16, and 18 (B–E traces in Figure 1).

#### 3.1.2. Glucoheptoamidated PAMAM G3–Cytisine Conjugates, G3^Cgh^

The conjugates bearing variable amounts of cytisine (**C**) were obtained after functionalization of **C** with (*p*-nitrophenyl)chloroformate (NPCF). **C** is an alkaloid originating from some plants [24] and is used to treat nicotine addiction [21]. It can be easily converted into *N*-acyl derivatives by reaction with acyl chlorides [25,26]. We have obtained the *N*-(*p*-nitrophenyl)carbonate derivative (**CNPC**) and characterized it by ^1^H and ^13^C-NMR spectroscopy. The 1-D and 2-D ^1^H COSY spectra of **CNPC** are presented in Figure 2. There are six methylene groups in **C**: 2, 4, 6, and 13 (see Scheme 1 for atom numbering). The largest magnetic nonequivalence occurs for geminal protons in 6-CH_2_ (δ_AB_ = 0.23 ppm), while 4-CH_2_ and 13-CH_2_ resonances are singlets. There is one set of **C** resonances in CDCl_3_ at room temperature. After replacement of N(3)-H with **NPC**, two sets of the ^1^H (Figure 2A) and ^13^C resonances (Figure 3) are observed, corresponding to *E*- and *Z*-conformers or chair(*syn*) and boat(*anti*) isomers similar to that for other *N*-acyl derivatives [25,26]. Due to the presence of the aromatic ring of the **NPC** substituent, the magnetic nonequivalences of 2-, 4-, and 6-CH_2_ protons are observed with ca. δ_AB2_ = 1.0 ppm, δ_AB4_ = 1.2 ppm, and δ_AB6_ = 0.5 ppm, respectively, for both isomers. The geminal methylene proton resonances were combined into pairs (AB-type spectra) based upon **a**, **b**, and **c** cross-peaks in COSY spectrum (Figure 2B). Additionally cross-peaks **d** between H-13 and H-1, **e** between H-5 and H-6, as well as cross-peaks **g** and **h** in aromatic region enabled full assignments within **C**, while two well-separated spectra of **NPC** substituents were combined into pairs of H-16 and H-17 doublets by cross-peaks **f** and **f′**. Two sets of resonances are apparent in the ^13^C-NMR spectrum (Figure 3A). The ^13^C resonances were assigned based upon 2-D ^1^H-^13^C HSQC and HMBC experiments (Figure 3B). The spectral assignments are presented in Table 1.

**NPCF** is convenient acyl chloride, which was used to substitute hydrogens of both the –OH and –NH groups. Furthermore, derivatized compounds like lauryl alcohol [7] or nimesulide [12,27] were reactive towards the amine groups of PAMAM dendrimers. The **CNPC** was used here to covalently attach **C** via carbonyl linker into G3 (see Scheme 1). Obtained G3-C conjugates were further derivatized with **GHL** to get aseries of G3^Cgh^ conjugates. Obtained conjugates were purified by extensive dialysis in order to remove DMSO and *p*-nitrophenol, the side product formed in the first step of synthesis. The conjugates were characterized by ^1^H-NMR spectroscopy. ^1^H-NMR spectra of obtained G3^Cgh^ conjugates are presented at Figure 4. Stoichiometry of the conjugates was determined by integration of aromatic **C** 4-H, 9-H, and 11-H as well as 13-CH_2_ (at 1.75 ppm) resonances versus b resonances of G3 core (with integral intensity correction due to overlapped C 5-H resonance). The number of gh substituents was determined as for G3^gh^ conjugates (Section 3.1.1.) with integrity adjustment due to overlap with **C** 5-H AB-type spectrum. Thus, the following conjugates of average stoichiometry were obtained: **G3^2C16gh^**, **G3^2C23gh^**, **G3^2C26gh^**, **G3^3C14gh^**, and **G3^9C11gh^**. These compounds were also characterized by IR spectroscopy and other methods (vide infra).

#### 3.1.3. Determination of Conjugate size by Dynamic Light Scattering (DLS)

Average size of conjugates was determined by the DLS method for ca. 0.1 mM solutions in 0.04 M acetate buffer pH 5 and for comparison in non-buffered aqueous solutions. The molecular sizes averaged by number of molecules (d(N)) and by volume (d(V)) are presented separately in Figure 5 and Figure 6 for the G3^gh^ and G3^Cgh^ conjugates, respectively. Unsubstituted G3 PAMAM dendrimer showed 3.6 nm diameter size, while substitution of G3 with gh resulted in increase of conjugate size until 1/3 of primary amine groups was substituted and decreased upon further substitution. However, fully substituted **G3^32gh^** showed the highest size of all G3^gh^ (for numbers, see Table 2). All volume-averaged values were slightly higher than number-averaged ones presumably due to aggregation of molecules which was considerably higher in non-buffered aqueous solutions (vide infra).

In the series of G3^2Cngh^, the size of almost completely substituted G3 core (2C, *n* = 26, and 4 non-substituted primary amine groups) was the biggest, while sizes of two other derivatives in the series did not change regularly. We concluded that size of conjugate depends not only on number of **C** and **gh** substituents but also on number of primary amine groups and on intermolecular association via hydrophobic interaction between **C**. The latter depended strongly on the number of **C** substituents. In the case of **G3^3C14gh^**, the volume-averaged size was considerably larger than the number-averaged one at pH 5. For **G3^9C11gh^** in this pH, the **G3^9C11gh^**-observed average size was above 70 nm with relatively low dispersion. In water, both of those conjugates as well as **G3^2C26gh^** and **G3^2C16gh^** showed also association with average diameter >70 nm, which dominantly contributed to a volume-averaged size (see Appendix A, Figure A1, and Table A1). Similar association was previously observed by AFM imaging for nimesulide-containing megamer G3^12G024Nim^, which showed 5.1 nm size measured by DLS for dilute solution, while average height of associates deposited on mica was 9 nm [12]. The molecular association also might be responsible for reported large size (185 nm) of PAMAM G5 measured by DLS [28]. In the case of the highly loaded cytisine conjugate, **G3^9C11gh^**, as well as in the mentioned nimesulide megamer, hydrophobic association between aromatic parts of C (and Nim) is responsible for association.

#### 3.1.4. ζ Potential and Electrophoretic Mobility

PAMAM G3 dendrimer as well as partially substituted G3^gh^ derivatives were expected to bind hydrogen cations both into internal tertiary and terminal primary nitrogen atoms. Therefore, we have examined the conjugate in buffered solutions at pH 5 and compared ζ potential of the conjugates in non-buffered solutions. The results are illustrated by graph in Figure 6.

In the series of G3^gh^ conjugates in pH 5, we have observed the decrease of ζ potential upon increasing levels of primary amine group substitution with gh, which is caused both by a decreasing number of protonated primary amine groups and a decreasing basicity of ternary amine nitrogen atoms upon glucoheptoamidation. The ζ potential values for G3^Cgh^ conjugates with 2C and 3C substituents are higher than expected only on the basis of primary amine group availability. The number of free primary amine groups for **G3^2C16gh^**, **G3^2C23gh^**, **G3^2C26gh^**, and **G3^3C14gh^** are 14, 7, 4, and 15, respectively. The non-cytisine containing analogues for **G3^2C16gh^** and **G3^3C14gh^** are **G3^16gh^** and **G3^18gh^** (with 14 and 16 free primary amine groups), respectively, which both showed considerably lower ζ potential than **G3^2C16gh^** and **G3^3C14gh^**. Instead, the **G3^2C23gh^** and **G3^2C26gh^** conjugates revealed ζ potential close to those half-filled **G3^16gh^** and **G3^18gh^** non-cytisine analogues.

Obviously, a substitution of primary amine groups with 2C and 3C resulted in increase of ternary nitrogen atoms basicity, enhanced protonation, and higher values of ζ potential for **G3^2C23gh^** and **G3^2C26gh^**. This rule holds true also for two low substituted cytisine conjugates, namely **G3^2C16gh^** and **G3^3C14gh^**.

Another reason for such behavior can be related to association of C-containing conjugates; in such a case, the contribution of highly positive associated cations may involve the ζ potential as it can be seen for **G3^9C11gh^**, for which considerable association takes place even at pH 5 as was evidenced by size measurement (vide supra).

Additionally, all **G3^Cgh^** conjugates showed higher values of ζ potential in water (for numbers, see Table A1) in comparison with all G3^gh^ analogues (except **G3^32gh^**) which indicated enhanced basicity of ternary nitrogen atoms in the former.

Although ζ potential values at pH 5 followed the dependence on number of primary nitrogen groups available for protonation, the electrophoretic mobility of all studied conjugates at pH 5 was the same due to a high charge of cations. Thus, we performed the PAGE electrophoresis using buffer pH 7.2. In this experiment, we observed different electrophoretic mobility of conjugates. The migration rate of dendrimers in gel was as follows (from the fastest to the slowest): **G3^F^** > **G3^5ghF^** > **G3^10ghF^** > **G3^16ghF^**. The migration rate of **G3^9C11ghF^** was close to **G3^10ghF^**, and the migration rate of **G3^2C16ghF^** was the same as that of **G3^16ghF^**. These two last observations confirmed the earlier hypothesis that electrophoretic mobility depends not only on a number of primary amine groups but also on the increase of cytisine-induced basicity of ternary nitrogen atoms (Table 2).

#### 3.1.5. Differential Scanning Calorimetry

DSC studies of both series of conjugates provided some information on internal dynamic behavior of the conjugates. Figure 7 and Figure 8 show the comparison of heat-flow rates versus temperature for **G3^gh^** and **G3^Cgh^** conjugates, respectively. The glass transition process as a jump of heat-flow rate in the so-called glass transition temperature (T_g_) was observed in all investigated samples of G3^gh^ and G3^Cgh^. DSC studies of both series of conjugates provided some information on internal dynamic behavior of the conjugates. We found that glass transition temperature (T_g_) depended on the number of gh substituents within G3^gh^ series, i.e., the lowest T_g_ was observed for naked G3 (−19.1 °C) and then continuously increased up to 25.9 °C for ca. half-substituted G3^18gh^ to finally drop down into 20.0 °C for G3^32gh^ (see Figure 7 and Table 2). It is well established that calculated change of heat capacity at glass transition temperature, ΔC_p_ (column 3 in Table 3), after conversion into molar change of heat capacity (column 5 in Table 3) enables close insight into molecule flexibility. In specific, each mobile unit in the chain of a polymer molecule contributes 11 J·K^−1^·mol^−1^ to the change of heat capacity at T_g_ [29,30,31,32]. The value of 11 J·K^−1^·mol^−1^ is an average value and depends on the bonds which are naturally different between different kinds of atoms. The ratio of measured heat capacity increment (ΔC_p_) to the contribution of a single mobile unit allows an estimate to be made on the total number of mobile units (beads) in an amorphous molecule which starts a mobile at the glass transition. According to this relationship, we calculated the number of beads within the conjugates. Thus, the calculated number of beads for naked G3 was 343–344, while the number of rotationally free bonds within G3 equals 334 if imposing the restriction on rotation within amide bond as well as (d)CH_2_-N bonds, leaving the free rotation of terminal (d_3_)CH_2_-NH_2_ bond. Despite a missing contribution (10 beads) which can be attributed to omitted bonds, an agreement between calculated and observed numbers is striking. Furthermore, every gh substituent contributes to flexibility of the conjugates. However, this contribution depends on the level of G3 substitution.

The experimentally obtained number of beads was compared with number of bonds for every conjugate (Figure 9). For low-substituted conjugates, i.e., **G3^5gh^**, **G3^10gh^**, and **G3^9C11gh^**, an observed number of beads was considerably lower than number of bonds. This suggests that attached gh moieties are not fully labile at glass transition temperature. The reasons for restricted rotation within those conjugates are not only high rotation barrier for *syn-anti* amide bond isomerization but also hydrogen bonds between gh-hydroxyl groups and surrounding primary amines or even folding of gh inside the PAMAM G3 core. Upon further substitution of G3 with gh and C moieties, the number of beads grows rapidly with a maximum at nearly half-filled dendrimers, i.e., **G3^16gh^**, **G3^3C14gh^**, and **G3^2C16gh^**. When more than half of dendrimer surface amine groups are substituted (**G3^18gh^**, **G3^2C23gh^**, and **G3^2C26gh^**), the number of beads becomes nearly equal to the number of bonds. Fully substituted **G3^32gh^** showed the highest number of beads of all gh-derivatized conjugates, which was consistent with a highest packing of the conjugate periphery. The local maximum for half-filled **G3^16gh^** suggested unrestricted conformational lability not only of gh chains but also of internal chains of G3 core.

#### 3.1.6. Toxicity of Conjugates

PAMAM dendrimers are considered one of the most promising nanoparticles used as vehicles for many drugs; however, the toxicity and immunogenicity of amine-terminated dendrimers are too high. We have chosen third-generation PAMAM dendrimers as the core of vehicles because it has relatively low toxicity; an optimal number of free primary amine groups; spheroidal, globular shape; and 3.6 nm diameter similar to proteins found in the human body (e.g., insulin) [33]. **G3** was substituted with **GHL** to convert stepwise 32 amine groups, and the conjugates were tested for cytotoxicity on normal human fibroblast cells (BJ). The results are presented in Figure 10. Neat **G3** reduced fibroblasts viability down to 65% at 3.1 µM and killed almost all cells at 12.5 µM concentration. Substitution of G3 PAMAM with 5, 10, or 16 GHL residues significantly elevated cell viability compared to naked **G3**. In the case of **G3^16gh^**, first, symptoms of cell damage were observed at 100 µM concentration but significant already at 200 µM concentration. These results are consistent with other reports which proved that cytotoxic action of PAMAM dendrimers correlated with the number of primary amino groups; their cytotoxicity was reduced when amine groups were substituted with acetyl groups or PEG tails [34]. Recently, we have used fully glucoheptoamidated G3 as a doxorubicine drug carrier [10,11]. We hypothesized that partial glucoheptoamidation may reduce amine group toxicity, retaining cationic character of a carrier. Thus, we have chosen the compromised G3^16gh^ and G3^11gh^ and attached additionally cytisine because of its low toxicity and common usage in medicine as antismoking drug [21,35]. Our preliminary studies indicated that cytisine was nontoxic not only against normal human fibroblasts (BJ) but also against human squamous cell carcinoma (SCC-15) and U-118 human glioma cells up to 500 µM after 24 h incubation (data not shown). However, an anticancer potential of cytisine against lung cancer cells [36] and HepG2 human hepatocellular carcinoma cells from few micromolar concentrations has been reported [37]. We found that **G3^2C16gh^** and **G3^9C11gh^** conjugates were not toxic up to 200 µM concentration and even stimulated 20% BJ cells growth at 3.1 and 50 µM concentrations, respectively (Figure 10).

Generally, the toxicity of nanoparticles depends not only on charge or charge-related ζ potential but also on size of nanoparticle. In non-phagocytic cells, small size of nanoparticles correlates with increased cytotoxicity [34]. It was also proven that phagocytic cells (THP-1 cells and human monocytes) were more resistant for 30–70 nm silica or 20–200 nm silver nanoparticles than for smaller nanoparticles [38,39]. Thus, generally, the aggregated nanoparticles are not as toxic as smaller concentrations of the same nanoparticles that fail to coalesce [40,41]. Here, we found that the high cytisine-loaded **G3^9C11gh^** conjugate was not toxic despite the presence of 20 free amine groups in the conjugate, while its non-cytisine analogue, **G3^10gh^**, reduced cell viability to 50% already at the 50 µM concentration. This may be due to high aggregation of **G3^9C11gh^** as evidenced by DLS measurements of hydrodynamic diameter, which were above 70 nm both in neutral and acidic conditions.

#### 3.1.7. Cellular Accumulation of Conjugates

Cytotoxicity of PAMAM dendrimers and other nanoparticles may be a consequence of their cellular absorption and accumulation efficiency. In this study, BJ normal human fibroblasts were incubated with FITC-labelled conjugates (**G3^F^**, **G3^5ghF^**, **G3^10ghF^**, **G3^16ghF^**, **G3^2C16ghF^**, and **G3^9C11ghF^**) for 1, 3, 5, or 24 h. The accumulation of FITC-labelled conjugates was determined quantitatively by fluorescence measurements and additionally evaluated by visual inspection based on images obtained with confocal microscope. We found that all compounds were efficiently absorbed and accumulated in BJ cells already from the first hour of incubation, but the most efficient uptake was observed in the cases of **G3^5ghF^** and **G3^10ghF^**. Slightly lower levels of **G3^16ghF^** and **G3^2C16ghF^** were noticed in all time points, but considerably lower accumulation of **G3^9C11ghF^** and **G3^F^** was detected especially after 24 h incubation (Figure 11 and Figure 12).

Dendrimers and PAMAM-derived conjugates may enter cells through multiple pathways, including macropinocytosis and clathrin- and caveolae-dependent endocytosis. Although we have not studied the cellular mechanisms of synthesized conjugates penetration, the confocal microscopy images showed that all absorbed conjugates remained in endocytic and in lysosomal vesicles, with a small part of the dendrimer present in the cytoplasm of the cells (Figure 12). The influx of studied compounds also can be executed by charge-mediated adsorptive endocytosis, since each conjugate has a non-substituted free amino groups, generating positive surface charge [13,42,43,44].

It cannot be excluded that the mechanism of phagocytosis plays a significant role in the degree of extracellular transport of studied compounds. Normal human fibroblasts belong to nonprofessional phagocytes, which may also undertake phagocytosis relatively less frequently [45]. We have anticipated that G3^F^ dendrimer will enter into BJ cells the most efficiently due to its the highest positive charge and the strongest interactions with negatively charged cellular membrane. Unfortunately, this probably caused disturbances of the membrane structure, reduced cell viability, and eventually lowered efficiency of its active uptake. Attachment from 5 to 16 GHL residues to G3 dendrimer increased significantly the uptake of **G3^5gh^**, **G3^10gh^**, and **G3^16gh^** conjugates after 24 h incubation compared to naked G3.

Substitution of amine groups by glucoheptoamide substituents maintaining the conjugates as very well soluble in water did not considerably enlarge the molecular size but could trigger additional paths for cellular influx. For instance, it was proven that hydroxyl-terminated G4 dendrimers were absorbed by primary microglia with other mechanisms including pinocytosis, caveolae, and aquaporin channels [46].

## 4. Conclusions

Stepwise glucoheptoamidation of primary amine groups of PAMAM G3 dendrimer enables tuning of physicochemical properties of the conjugates, influencing their molecular size and ζ potential. The local maximum of size and intermediate ζ potential in water (24 mV) were achieved for *n* = 10. Further substitution (*n* = 16) results in decrease of both parameters. All glucoheptoamidated G3 conjugates have glass transition temperature (T_g_) below 26 °C. Glucoheptoamide substituents show immobilization near T_g_ in low substituted conjugates (*n* = 5 and 10) probably due to hydrogen bonding between polyhydroxylated substituents and the remaining primary amine groups of G3 or due to folding of glucoheptoamide chains inside the interior of G3. All conjugates accumulate fast in normal fibroblast cells (BJ). Low-substituted conjugates exert toxicity against BJ, diminished upon increasing levels of glucoheptoamidation, with the least for the conjugate with *n* = 16 and 16 free amine groups.

Such conjugates containing covalently attached two cytisine equivalents were then tested as carriers for hydrophobic, biologically innocent alkaloid-cytisine; for accumulation; and for toxicity on BJ cells. The conjugate of 4.2 nm size and ζ potential 7 mV in pH 5 and 15 mV in water accumulates efficiently in BJ cells, is mostly located in endosomes and lysosomal vesicles after 24 h, and is not toxic for BJ cells up to 200 µM concentration. Another cytisine-loaded conjugate containing 9 cytisine residues associates in diluted aqueous solution, reveals 30 mV ζ potential, accumulates slower in BJ cells, and is not toxic.

We conclude that an appropriate PAMAM-based drug carrier for blood administration should have ca. 50% of amine groups blocked and should be used to covalently bind a drug, targeting molecule and solubilizing substituents like glucoheptoamide applied here. A special care must be taken on substitution with hydrophobic drug molecules to avoid association of conjugates.

Half-glucoheptoamidated G3 dendrimer shows low diameter in water (ca. 2 nm) and high surface charge density (zeta potential ca. 10 mV) and efficiently accumulates in normal human fibroblast cells. In a lysosomal pH 5, this molecule expands to 4.6 nm size accompanied by decrease of zeta potential to ca. 1 mV. Both effects may enable lysosomal release of a potential drug from the conjugate. The flexibility of drug conjugate seems to be also an advantageous property, which can be recognized by glass transition temperature below 30 °C.

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
