# Peer review of "Stepwise Glucoheptoamidation of Poly(Amidoamine) Dendrimer G3 to Tune Physicochemical Properties of the Potential Drug Carrier: In Vitro Tests for Cytisine Conjugates"

_pharmaceutics, 2020, doi:10.3390/pharmaceutics12050473_

Round 1

Reviewer 1 Report

The study by Czerniecka-Kubicka et al. focus on the third generation of poly(amidoamine) (PAMAM) dendrimers and their chemical modification with D-glucoheptono-1,4-lactone, cytisine and  glucoheptoamide substituents with the aim to propose a novel nanoparticle-based drug delivery platform. The Authors characterised physico-chemical properties and behaviour of the conjugates using various experimental techniques and made an attempt to link them to the results from biological test obtained on the  BJ cell line of normal human fibroblasts. In the experiments on the BJ cell line the Authors have focused on the third generation of PAMAM dendrimer as a core of vehicle, and justified this its relatively low toxicity, an optimal number of free primary amine groups, spheroidal, globular shape, and the nanoscale diameter similar to proteins found in the human body. They found that the selected cytisine-modified PAMAM dendrimers were non-toxic against BJ, but that they likely did not exert sufficient selectivity (effect on malignant cell lines). Based on a broad range of methods and experimental results the Authors propose possible and feasible ways to optimisation of the nanoparticles which might make them suitable for further validation studies.

From the general point of view, the central idea, focus and purpose are important and the conception reasonably grounded. The subject of this work is novel, interesting and fits within the scope of the journal Pharmaceutics.

In my opinion, it is a nice piece of work and I do not have any serious objections to the present version. There are, however, a few minor points, which still require some attention.

General comments

Abstract lacks a clear conclusion, which would link the main result/product to the future endeavours (compare with the recommendations in the Conclusion part).

I appreciate the broad spectrum of experimental techniques used by the Authors (including calorimetric studies and visualisation). Nevertheless, the approach merging chemistry with biophysics and biology has also some drawbacks - the text of the manuscript is too “dense” with many abbreviations, which makes it difficult to follow the story. (The same hold true for Figure 10 presentation.)

Specific comments

  • The sentence in Abstract: ”None- or low-substituted dendrimers accumulated very fast in fibroblast cells, although they were toxic already at 10 µM concentration.” is difficult to comprehend.
  • The Author might reconsider the choice of keywords. Why “size”? Or why “confocal microscopy”? How about the other experimental techniques? Moreover, size and zeta potential = DLS (dynamic light scattering) technique.
  • The last sentence of the Introduction should be moved to a more appropriate part (The synthetic path of the conjugates and atom numbering are presented at Scheme 1.)
  • In 2.2.1. PAMAM G3 substituted with glucoheptonamide part: “3 ml dmso each,” Why not in capital letters? DMSO?
  • Use a non-breaking space esp. between a number and a unit of measurement.
  • 10 °C not 10°C
  • 4 % not 0.4% (the latter is used as adjective). Correct throughout the text.
  • 2.3. Conjugates labeled with FITC were fluorescent-labeled….
  • FITC should be introduced in the above section, not in 2.3.3.
  • Confocal microscopy visualization: (10 minutes, RT) RT is a very vague information.
  • 3.4. Statistical analysis: “the differences between treated and non-treated control… a statistical analysis was performed using the non-parametric Kruskal–Wallis test.” What was the reason for using this test that is a nonparametric alternative for one-factor analysis of variance? I see two factors in question, type of dendrimer and concentration categories for measured viability. Actually, the nature of the latter factor allows for using regression analysis.
  • “The index of probability of 0.05 or less (P < 0.05) was considered significant in the comparative analysis.” This is not an index (!) This is probability, more specifically, the achieved level of significance. Figure 10: I assume that the comparisons were not corrected for type-one error inflation due to multiple tests performed (probably not needed for an exploratory-type of study, but it should be indicated).
  • The detected zeta potential values along with the recommendation in the Conclusion” (Preferably the potential of such drug conjugate should be slightly positive (< 15 mV)) point to the range of potential instability of the system. Could you comment on this?
  • In the part 3.1.7. Cellular accumulation of conjugates: “The accumulation of FITC-labelled conjugates was determined quantitatively by fluorescence measurements and visualized with confocal microscope….Slightly lower level of G316ghF, G32C16ghF, and significantly lower accumulation of G39C11ghF and G3F was noticed (Figure 11).”    What was the latter statement based on the visual inspection of the field only? With separate experiments on the quantitative evaluation of the fluorescence?

Author Response

Dear Sir (Madam),

Please find the introduced changes printed in red and our answers in blue.

Thank you very much for detailed wotk on our manuscript,

With our best regards,

Authors

Reviewer 1

General comments

Abstract lacks a clear conclusion, which would link the main result/product to the future endeavours (compare with the recommendations in the Conclusion part).

We have changed slightly the Conclusion part and addressed the last sentence of Abstract to revised Conlusions by adding the following sentence:

The half-glucoheptoamidated PAMAM dendrimer was selected as promising anticancer drug carrier for further applications.

I appreciate the broad spectrum of experimental techniques used by the Authors (including calorimetric studies and visualisation). Nevertheless, the approach merging chemistry with biophysics and biology has also some drawbacks - the text of the manuscript is too “dense” with many abbreviations, which makes it difficult to follow the story. (The same hold true for Figure 10 presentation.)

Response to comment:

We have synthesized and tested many glucoheptonoamidated polyamidoamine conjugates and additionally cytisine-containing conjugates. Therefore to avoid descriptive or simple numbering of compound we have abbreviated them as G3Cgh, with upper index indicating how many substituents a conjugate contains. The names of compound are typed in bold in order to make them clearly recognizable for readers. This also include some abbreviations of D-glukoheptono-1,4-lactone (GHL), cytysine (C ), p-nitrophenylchloroformate (NPCF), and cytisine p-nitrophenylcarbonate (CNPC). All these names are in capital letters, while glucoheptonoamide substituents are mentioned with small letters gh.

The abbreviations of methods used by us are common; dynamic light scattering (DLS) and Differential Scanning Calorimetry (DSC), nuclear magnetic resonance spectroscopy (NMR) together with 2-dimensional homo- and heteronuclear COSY, HSQC, and HMBC experiments are regular spectroscopic tools used by chemists and we would prefer not to specify the experimental details.

We realize that the paper is difficult to read due to contribution of many methods. Nonetheless, it will probably find many readers, especially after changes in keywords, which we have introduced. We have encountered the methods in keywords in order to make the paper easily visible for readers specialized in DSC and DLS. The results obtained by these two techniques are textbook elegant.

Nonetheless, in the revised version we have shortened the DSC discussion in order to make it more readable.

In our opinion Figure 10 is necessary for the reader to understand very significant differences in the toxicity profiles of the compounds tested. Abbreviations of tested compounds are consistent with the text throughout the paper.

Specific comments

  • The sentence in Abstract: ”None- or low-substituted dendrimers accumulated very fast in fibroblast cells, although they were toxic already at 10 µM concentration.” is difficult to comprehend.

Response: The sentence has been clarified as follows:

Native or low substituted dendrimer conjugates accumulated efficiently in fibroblast cells at non-toxic 1 µM concentration.

  • The Author might reconsider the choice of keywords. Why “size”? Or why “confocal microscopy”? How about the other experimental techniques? Moreover, size and zeta potential = DLS (dynamic light scattering) technique.

Response:

Thank you very much for the remarks. We have removed confocal microscopy, which seems obvious and replaced it with BJ normal human fibroblasts. We have replaced size and zeta potential with dynamic light scattering and also added differential scanning calorimetry, which is not very often used method to study PAMAM-based derivatives.

  • The last sentence of the Introduction should be moved to a more appropriate part (The synthetic path of the conjugates and atom numbering are presented at Scheme 1.)

Response:

We have removed the sentence from an Introduction. It was placed at the beginning of Results and discussion section and remained as such.

  • In 2.2.1. PAMAM G3 substituted with glucoheptonamide part: “3 ml dmso each,” Why not in capital letters? DMSO?

Response:

Yes, we have corrected the abbreviation to keep naming of compounds consistent.

  • Use a non-breaking space esp. between a number and a unit of measurement.
  • 10 °C not 10°C
  • 4 % not 4% (the latter is used as adjective). Correct throughout the text.

Response:

We have corrected these typos throughout the text. Thank you very much for careful examining our manuscript.

  • 3. Conjugates labeled with FITC were fluorescent-labeled….

Response:

We have corrected the text as follows (section 2.2.3):

The selected conjugates were fluorescent labeled by stepwise addition of fluorescein isothiocyanate (FITC) dissolved in ethanol into typically 10 µmoles of following conjugates: G3, G35gh, G310gh, G316gh, and G39C11gh and G32C16gh in methanol to obtain G3F, G35ghF, G310ghF, G316ghF, and G39C11ghF, and G3F for biological tests and electrophoretic experiments. Each conjugate molecule contained an average one FITC residue.

  • FITC should be introduced in the above section, not in 2.3.3.

Response:

Both types of conjugates: cytisine-free G3gh (syntheses described in section 3.2.1.) and cytisine-containing glucoheptoamidated G3Cgh ones (syntheses described in section 3.2.2.) were labeled with FITC. Therefore it cannot be joined with 3.2.1. and 3.2.2. Although labeling proteins and other amine-containing compounds is common procedure, we have applied the solution of FITC in ethanol and this protocol is valuable for other researches. We insist to remain it described in separate section.

  • Confocal microscopy visualization: (10 minutes, RT) RT is a very vague information.

Response:

We have specified this in the revised version as follows (section 2.3.3.):

(for 10 minutes at room temperature).

  • 4. Statistical analysis: “the differences between treated and non-treated control… a statistical analysis was performed using the non-parametric Kruskal–Wallis test.” What was the reason for using this test that is a nonparametric alternative for one-factor analysis of variance? I see two factors in question, type of dendrimer and concentration categories for measured viability. Actually, the nature of the latter factor allows for using regression analysis.

Response:

We have used non-parametric Kruskal-Wallis test due to lack of normal data distribution in each investigated group (n = 9, triplicate from three independent experiments). In this situation one-factor analysis of variance is not correct. We might use regression analysis, but in our the best knowledge non-parametric test is the most appropriate. Our goal was to understand what concentration of each compound significantly decreased cell viability, not just to compare relationships with each other. In our previous papers and studies, we have been using non-parametric tests for years, which have always been recognized as the right way to evaluate the obtained results.

  • “The index of probability of 0.05 or less (P < 0.05) was considered significant in the comparative analysis.” This is not an index (!) This is probability, more specifically, the achieved level of significance.

Response:

Thank you for the valuable amendment, an error was corrected as follows (section 2.3.4.):

For the cell culture assays, to estimate the differences between treated and non-treated control samples, a statistical analysis was performed using the non-parametric Kruskal–Wallis test. P< 0.05 was considered as statistically significant. Calculations were performed using Statistica 13.3 software (StatSoft).

  • Figure 10: I assume that the comparisons were not corrected for type-one error inflation due to multiple tests performed (probably not needed for an exploratory-type of study, but it should be indicated).

Response:

Yes, also in our opinion correction of type-one error inflation is not required for exploratory type of study. In our statistical analysis we have used non-parametric Kruskal-Wallis test and post-hoc multiple comparisons of mean ranks for all groups (Statistica, Stat-Soft). This test belongs to weak statistical tests. Therefore, obtained P value is a reliable value. If we used a strong, parametric test (ANOVA) the probability of making the first error in many comparisons would be really high and Bonferroni correction would be necessary. In case of non-parametric Kruskal-Wallis the obtained P values are subject to a marginal error; therefore we do not apply type-one error inflation.

  • The detected zeta potential values along with the recommendation in the Conclusion” (Preferably the potential of such drug conjugate should be slightly positive (< 15 mV)) point to the range of potential instability of the system. Could you comment on this?

It is difficult question for us to answer. At the beginning we determined zeta potential for all conjugates at pH 5 because we plan to use the conjugates to carry anticancer drug. As we have found by confocal microscopy in previous studies other tested drug-conjugates were entering the cells on endocytic way and finally were “digested” in lysosomes. Therefore we used buffered pH 5 solution to determine zeta potential of tested carriers here. Then we realized that obtained value depends on the kind and concentration of buffer. Because cell membrane affinity of positively charged carrier is crucial and that event occurs at pH nearly 7 (pH of culture media, optimal for cell culture) we have also studied the zeta potential in non-buffered solution and found it surprisingly equal near zero for naked G3, although other values were reported in 10 mM NaCl solution, 43.3

[M. A. Dobrovolskaia, A. K. Patri, J. Simak, J. B. Hall, J. Semberova, S. H. De Paoli Lacerda, S. E. McNeil. Nanoparticle size and surface charge determine effects of PAMAM dendrimers on human platelets in vitro. Mol Pharm. 2012 March 5; 9(3): 382–393. doi:10.1021/mp200463e]

and ca 5.8 mV in water.

[Y. Zeng, Y. Kurokawa, T.-T. Win-Shwe, Q. Zeng, S. Hirano,  Zh. Zhang, H. Sone. Effects of PAMAM dendrimers with various surface functional groups and multiple generations on cytotoxicity and neuronal differentiation using human neural progenitor cell. J. Toxicol. Sci. Vol.41, No.3, 351-370, 2016].

Furthermore, in case of PAMAM G4 the value of zeta potential was ca 22 mV, which increased considerably upon 50 % substitution with C12 (C = 2-hydroxydodecyl substituent, without size disturbance; both 4.5 nm diameter). This observation corroborate well with our results.

For other conjugates we found the increase of zeta potential from low in pH 5 to ca 14 mV or higher for G310gh, G332gh and all cytisine conjugates. It seems to us that positive zeta potentials observed for G310gh, G332gh and all cytisine conjugates enable the synthesized conjugates to bind to cell membrane with high affinity.

Sorry, we do not know what kind of potential instability you meant and are unable to comment this remark. Nevetheless, we have introduced some changes into Conclusion part as follows:

We conclude that an appropriate PAMAM-based drug carrier for blood administration should have ca 50 % of amine groups blocked and used to covalently bind a drug, targeting molecule and solubilizing substituent like glucoheptoamide applied here. A special care must be taken on substitution with hydrophobic drug molecules to avoid association of conjugates. Half-glucoheptoamidated G3 dendrimer shows low diameter in water (ca 2 nm) and high surface charge density (zeta potential ca 10 mV) and efficiently accumulate in normal human fibroblast cells. In a lysosomal pH 5 this molecule expands to 4.6 nm size accompanied by decrease of zeta potential to ca 1 mV. Both effects may enable lysosomal release of potential drug from the conjugate. The flexibility of drug conjugate seems to be also advantageous property, which can be recognized by glass transition temperature below 30 ℃.

  • In the part 3.1.7. Cellular accumulation of conjugates: “The accumulation of FITC-labelled conjugates was determined quantitatively by fluorescence measurements and visualized with confocal microscope….Slightly lower level of G316ghF, G32C16ghF, and significantly lower accumulation of G39C11ghF and G3F was noticed (Figure 11).”    What was the latter statement based on the visual inspection of the field only? With separate experiments on the quantitative evaluation of the fluorescence?

Response:

Our conclusion was based on both quantitative evaluation of the fluorescence (Figure 11) and also on visual inspection form confocal microscopy (Figure 12). As you can see on Figure 11 the most efficient accumulation was observed in case of G35ghF and G310ghF at each time point. Accumulation of G316ghF, G32C16ghF was weaker at each time point and G39C11ghF and G3F especially after 24 h incubation. Therefore we have changed above sentence to:

The accumulation of FITC-labelled conjugates was determined quantitatively by fluorescence measurements and additionally evaluated by visual inspection based on images obtained with confocal microscope. We found that all compounds were efficiently absorbed and accumulated in BJ cells already from first hour of incubation, but the most efficient uptake was observed in case of G35ghF and G310ghF. Slightly lower level of G316ghF, G32C16ghF was noticed in all time points, but considerably lower accumulation of G39C11ghF and G3F was detected especially after 24 h incubation (Figure 11 and 12).

Thank you for your valuable advice.

Reviewer 2 Report

The paper describes the use of a PAMAM G3 dendrimer functionalized by a variable number of glucoheptoamide derivatives and cytisine, an alkaloid used in anti-nicotinic therapy. The different compounds are analysed by a lot of different techniques, and the cytotoxicity was measured. All the work is well carried out, but the main question is for what? The idea is to have a drug carrier, but only a very low number of drugs have been conjugated to a large dendrimer, no experience of release has been attempted, and in view of the type of linker, the release is not possible. Furthermore, the choice of the substance to be carried is not justified, and questionable. For these reasons, I don’t think that is paper is suitable for Pharmaceutics.

However, I have reviewed thoroughly the paper, and here are my comments:

  • In the introduction, in the description of the criteria needed for a suitable drug carrier, a 6th criteria is missing: the efficiency of the release of the drug. In many cases, when a drug is conjugated to a dendrimer, its efficiency is lost.
  • Figures 1, 2, 3 and the Table 1 should be in the Annex (or Supporting Information)
  • The reason for the possible associations of dendrimers in some cases, as shown by DLS is not given. One may anticipate interaction of the external C=O of cytisine with the remaining NH2 of another dendrimer. To demonstrate or discard this possibility, a dendrimer without any NH2 should have been synthesized (only glucoheptoamide derivatives and cytisine).
  • For DSC, it is written that it gives “some information on the internal dynamic behavior”. Yes, but not only. The most dynamic part of dendrimers is always the surface and close to the surface, and this is what is demonstrated here, since the functionalization of the surface induces large differences in the DSC results.
  • I don’t understand the use of beads in the DSC part. This DSC part is ‘indigestible”.
  • In the DSC part, the connection between the Tg and the temperature used for MNR experiments is false. The Tg is measured in bulk, the NMR is solution. It is not correct to speak about the cis-trans amide bond, it should be syn anti.
  • There are two ‘abnormal” results in the DSC results, concerning G3 9C11gh and G3 2C16gh, as can be seen in particular in the Table 3 and Figure 9, but no explanation is given.
  • I have not seen how many FITC have been conjugated per dendrimer.
  • In the conclusion, I disagree with the recommendation concerning the low drug payload. This is what has been observed in this study, but it cannot be a recommendation. How is it possible to expect any practical use of a large dendrimer to which only two active substances have been conjugated?

Author Response

Dear Sir (Madam),

Please find the introduced changes printed in red and our answers in blue.

Thank you very much for detailed wotk on our manuscript,

With our best regards,

Authors

Reviewer 2

All the work is well carried out, but the main question is for what? The idea is to have a drug carrier, but only a very low number of drugs have been conjugated to a large dendrimer, no experience of release has been attempted, and in view of the type of linker, the release is not possible. Furthermore, the choice of the substance to be carried is not justified, and questionable. For these reasons, I don’t think that is paper is suitable for Pharmaceutics.

Response:

Thank you for critical review.
You are absolutely right about urea link used to covalently attach a cytisine molecule; it is not cleavable.

However, the paper is not about cytisine, it is about a carrier, PAMAM G3 modified to obtain water-soluble, cationic carrier, which can be used to attach an anticancer drug. Recently we used naked G3 as carrier for nimesulide [12,27] and then celecoxib and F-Moc-L-Leu (all through amide linkers) and also fully G3 glucoheptonoamidated G3 to carry covalently doxorubicin linked via ester bond [11], and realized that a carrier can be modified in such a way, that it bears both amine groups and hydroxyl groups of gluconoheptoamide (or another polyhydroxyl substituent, like glycidol-derived or PEG). Amine groups must be partially removed from the carrier in order to reduce its toxicity, therefore we performed systematic derivatization to find compromised carrier.

We preliminarily studied biological activity of cytisine and found it not active on series of human cell lines, both normal and cancer. In these studies we could not find evidence the cytisine enters cell. We studied cytisine conjugates, which were labeled and in this way we could visualize if the cytisine conjugate enters the cells and also find out if it is cytotoxic. Fortunately it was not and we could use cytisine as model hydrophobic drug without impairment of cell.

However, I have reviewed thoroughly the paper, and here are my comments:

  • In the introduction, in the description of the criteria needed for a suitable drug carrier, a 6th criteria is missing: the efficiency of the release of the drug. In many cases, when a drug is conjugated to a dendrimer, its efficiency is lost.

Response:

Yes, we realize that release is another criterion. We have added this into Introduction as follows:

Such conjugates must fulfill the following criteria: (i) high affinity to cell membrane, (ii) good membrane permeation, (iii) water solubility for intravenous administration, (iv) selective binding to cancer cell membrane, (v) low systemic toxicity, (vi) prolonged blood circulation half-time, and (vii) ability to release a drug from a conjugate. The latter criterion requires cleavable linker. PAMAM dendrimer-based conjugates are good candidates as multidrug carriers to challenge against multidrug resistant cancers [19,20].

Thank you for your suggestion.

In fact we also know that in some cases drug molecule can stay bound with nanosized carrier and still exert biological effect, similar to that of “free” drug. We found it in case of doxorubicin linked to fully glucoheptoamidated PAMAM G3 that it killed glioblastoma cells four times more efficiently than DOX itself. In case of that conjugate (4 DOX per one dendrimer molecule) we found marginal release of DOX (0.1 % within 1 week) [11].

So, we studied toxicity of cytisine-bearing carrier, though cytisine itself was not toxic. We performed the measurements and found conjugate not toxic, although cell accumulation was evidenced.

  • Figures 1, 2, 3 and the Table 1 should be in the Annex (or Supporting Information)

Response:

The paper is mostly on tailoring the carrier, which is chemist’s duty working for medical application. The synthesized compounds were characterized by NMR spectroscopy and spectral details are important for chemists, which are also readers of Pharmacutics journal. We would prefer to leave the spectral cakes for them to watch without jumping to the end of paper. Please let us leave figure 1-3 in the place they are. The readers which are not familiar with regular chemistry can easily skip these three pages as well as Table 1 documentary.

  • The reason for the possible associations of dendrimers in some cases, as shown by DLS is not given. One may anticipate interaction of the external C=O of cytisine with the remaining NH2 of another dendrimer. To demonstrate or discard this possibility, a dendrimer without any NH2 should have been synthesized (only glucoheptoamide derivatives and cytisine).

We did not speculate on specific interaction because such statements would require evidences. However we have observed the association of hydrophobic drug substituted PAMAM dendrimers in case of Nimesulide attached to G3 as well as megameric G3-G0-Nimesulide conjugate by atomic force microscopy and DLS.

We have added the sentence on association by hydrophobic interaction though aromatic part of cytisine molecules as follows (page 14, above Figure 5):

In case of highly loaded cytisine conjugate, G39C11gh as well as in mentioned Nimesulide megamer rather hydrophobic association between aromatic parts of C (and Nim) is responsible for association.

Considering the synthesis of totally substituted G3 with cytisine and gh moieties we did that as far as possible and obtained not fully-substituted G32C26gh; the remaining average 4 amine groups were resistant to GHL reaction, presumably due to steric hindrance imposed by cytisine occupying next amine-terminated arm. Similar behavior we found in case of glycidol addition into G3 substituted with one FITC despite large glycidol excess used in a synthesis.

For DSC, it is written that it gives “some information on the internal dynamic behavior”. Yes, but not only. The most dynamic part of dendrimers is always the surface and close to the surface, and this is what is demonstrated here, since the functionalization of the surface induces large differences in the DSC results.

  • I don’t understand the use of beads in the DSC part. This DSC part is ‘indigestible”.

Response:

We have modified and shortened the discussion of this part, focusing on Figure 9 comparison between number of bonds and observed number of beads (“labile” bonds at glass transition temperature), as follows (page 17):

The experimentally obtained number of beads was compared with number of bonds for every conjugate (Figure 9). For low-substituted conjugates, i.e. G35gh, G310gh, and G39C11gh, an observed number of beads was considerably lower than number of bonds. This suggests that attached gh moieties are not fully labile at glass transition temperature. The reasons for restricted rotation within those conjugates are not only high rotation barrier for syn-anti amide bond isomerization but also hydrogen bonds between gh-hydroxyl groups and surrounding primary amines or even folding of gh inside the PAMAM G3 core. Upon further substitution of G3 with gh and C moieties the number of beads grow rapidly with a maximum at nearly half-filled dendrimers, i.e. G316gh, G33C14gh, and G32C16gh. When more than half of dendrimer surface amine groups are substituted (G318gh, G32C23gh, and G32C26gh) the number of beads becomes nearly equal to number of bonds. Fully substituted G332gh showed highest number of beads of all gh-derivatized conjugates, which was consistent with a highest packing of the conjugate periphery. The local maximum for half-filled G316gh suggested unrestricted conformational lability not  only gh chains but also internal chains of G3 core.

  • In the DSC part, the connection between the Tg and the temperature used for MNR experiments is false. The Tg is measured in bulk, the NMR is solution. It is not correct to speak about the cis-trans amide bond, it should be syn anti.

Response:

We removed the comment addressing NMR spectra. In fact we did not perform temperature dependent NMR measurements to observe coalescence temperature experimentally.

Yes, we have replaced cis-trans with syn-anti nomenclature throughout the entire paper.

  • There are two ‘abnormal” results in the DSC results, concerning G3 9C11gh and G3 2C16gh, as can be seen in particular in the Table 3 and Figure 9, but no explanation is given.

Response:

At Figure 9 we found one point placed in wrong place (G32C26gh, wrong x coordinate) in primary submission. We have corrected it and improved quality of Figure 9. We have shortly discussed DSC results in the revised version to make it digestable and more clear.

  • I have not seen how many FITC have been conjugated per dendrimer.

Response:

The average number of FITC attached was one label per macromolecule. The relevant fragment has been modified in section 2.2.3. as follows:

The selected conjugates were fluorescent labeled by stepwise addition of fluorescein isothiocyanate (FITC) dissolved in ethanol into typically 10 µmoles of following conjugates: G3, G35gh, G310gh, G316gh, and G39C11gh and G32C16gh in methanol to obtain G3F, G35ghF, G310ghF, G316ghF, and G39C11ghF, and G3F for biological tests and electrophoretic experiments. Each conjugate molecule contained an average one FITC residue.

  • In the conclusion, I disagree with the recommendation concerning the low drug payload. This is what has been observed in this study, but it cannot be a recommendation. How is it possible to expect any practical use of a large dendrimer to which only two active substances have been conjugated?

Response:

Yes, we meant only the case, although we have synthesized couple of high-loaded PAMAM dendrimers with celecoxib (and decorated with targeting folic acid and decorated with biotin) and nimesulid (and folic acid) and all of them became water insoluble and useless as drug delivery systems.

We have weakened our conclusion regarding “recommendation” in the revised version as follows:

We conclude that an appropriate PAMAM-based drug carrier for blood administration should have ca 50 % of amine groups blocked and used to covalently bind a drug, targeting molecule like biotin and solubilizing substituent like glucoheptoamide applied here. A special care must be taken on substitution with hydrophobic drug molecules to avoid association of conjugates. Half-glucoheptoamidated G3 dendrimer shows low diameter in water (ca 2 nm) and high surface charge density (zeta potential ca 10 mV) and efficiently accumulate in normal human fibroblast cells. In a lysosomal pH 5 this molecule expands to 4.6 nm size accompanied by decrease of zeta potential to ca 1 mV. Both effects may enable lysosomal release of potential drug from the conjugate. The flexibility of drug conjugate seems to be also advantageous property, which can be recognized by glass transition temperature below 30 ℃.

Reviewer 3 Report

The study presented by the authors discusses the functionalization of a third-generation PAMAM dendrimer by stepwise amidation with D-glucoheptono-1,4-lactone to obtain a carrier to cytisine that is used to treat nicotine addiction.

The structural characterization of all the systems is extensive and many techniques have been used to support the authors' proposal.  All the discussion on characterization is adequate and clear.

The assays performed to evaluate the cytotoxicity for therapeutic use are adequate and as expected the functionalization of the dendrimer with D-glucoheptono-1,4-lactone reduces the cytotoxicity by decreasing the number of positive surface charges. Also, it seems that the partial functionalization allows the system to remain soluble in the medium.

So I recommend the publication with minor revisions and that are those detailed below,

-in my opinion, the introduction should mention what the reason is for choosing with D-glucoheptono-1,4-lactone to functionalize the dendritic system, what advantages it has.

In the conclusions I think it is not entirely correct to opt for the fifth generation dendrimer without providing any data, the gap of generations from three to five is great synthetically effort, and why a fifth and not a fourth-generation? if the authors have any evidence should give some answer to this.

Author Response

Dear Sir (Madam),

Please find the introduced changes printed in red and our answers in blue.

Thank you very much for detailed wotk on our manuscript,

With our best regards,

Authors

Reviewer 3

In my opinion, the introduction should mention what the reason is for choosing with D-glucoheptono-1,4-lactone to functionalize the dendritic system, what advantages it has.

Response:

Thank you for getting this point. Before using D-glucoheptono-1,4-lactone we explored glycidol as terminal polydroxy substituent. However biological activity of R- and S-glycidol on dendrimer periphery needs some biological tests, which is our ongoing project. On the other hand common PEG-ylation results in obtaining products of high polydispersion of molecular weight (and molecular size). Monosaccharide lactones provide the opportunity to obtain monodisperse PAMAM-based polyhydroxylated carrier. We have explained this choice in Introduction by adding an explanation of the reagent choice as follows:

Such substitution allows to provide six hydroxyl group per one glucoheptoamide substituent and keep low molecular weight dispersity in contrary to commonly used PEG-ylation [4] as well as maintain very good water solubility of modified PAMAM [9-11]. 

In the conclusions I think it is not entirely correct to opt for the fifth generation dendrimer without providing any data, the gap of generations from three to five is great synthetically effort, and why a fifth and not a fourth-generation? if the authors have any evidence should give some answer to this.

Response:

You are right, it was general remark, based on opinions. In accordance with literature data the PAMAM G5 molecular size is 5.5 nm. The glomerular filtration is efficient up to macromolecules sized below 5 nm diameter. Thus, usually G5 derivatives are used in biological tests in vivo. The blood half-live time for G5 is considerable longer than for that for analogous G3 and G4 systems. However, we did not study our conjugates on animals. We still test the carriers and anticancer drug – loaded conjugates to find the best system to kill cancer cells in vitro. It is much easier to synthesize G3 PAMAM dendrimer of high purity and therefore we use G3 and test its conjugates in vitro. Commercially available PAMAM G5 is alternative solution to scale-up some remarkably active drug-delivery system we search for.

We have removed this suggestion from the revised version of our paper.

Round 2

Reviewer 2 Report

The paper has been improved.